## Research Article

mental health; ADHD; children; Africa; Strength and Difficulties Questionnaire; Tests of Variables of Attention; psychosocial adjustment; caregiving quality; Benin; poverty

**Corresponding author:**
Michael J. Boivin;
Email: boivin@msu.edu

# Associations between the Strengths and Difficulties Questionnaire (SDQ) and Tests of Variables of Attention (TOVA) in rural school-aged children in Benin Africa

Roméo Zoumenou[1,2], Nathalie Costet[3], Michael J. Boivin[4,5,6] (iD), Jaqueline Wendland[2] and Florence Bodeau-Livinec[3]

[1]Paris Descartes' Cognition, Conduct and Human Behavior Doctoral School (ED 261), Université Paris Cité, Paris, France; [2]Laboratoire Psychopathologie et Processus en Santé, Institute de Psychologie, Boulogne, France; [3]EHESP, Inserm, IRSET (Institut de recherche en santé, Environnement et travail) – UMR_S 1085, University de Rennes, Rennes, France; [4]Department of Psychiatry, Michigan State University, East Lansing, MI, USA; [5]Department of Neurology & Ophthalmology, Michigan State University, East Lansing, MI, USA and [6]Department of Psychiatry, University of Michigan, Ann Arbor, MI, USA

## Abstract

Sub-Sahara Africa (SSA) children are at high-risk neurodevelopmentally due to the prevalence of infectious disease, nutritional deficiencies and compromised caregiving. However, few mental health screening measures are readily available for general use. The Strengths and Difficulties Questionnaire (SDQ) has been used as a mental health screening measure in the SSA, but its psychometric properties are not well understood. Five hundred and sixty-six mothers completed the SDQ for their 6-year-old children in rural Benin north of Cotonou. These were mothers who had been part of a malarial and intestinal parasite treatment program and micronutrient fortification intervention program during pregnancy for these children. Their study children ($N$ = 519) completed the computerized Tests of Variables of Attention (TOVA-visual) as a performance-based screening assessment of attention deficit and hyperactivity disorders. In evaluating the relationship between the SDQ and TOVA, we controlled for maternal risk factors such as depression, poor socioeconomic status and educational level, along with the child's schooling status. TOVA measures of impulsivity were significantly related to SDQ emotional and hyperactivity/inattention difficulties. TOVA inattention was related to SDQ emotional difficulties. The triangulation of maternal risk factors (e.g., depression), the SDQ and the TOVA can provide effective screening for mental health issues in SSA children.

## Social media summary

Caregiver reports of their child's emotional and behavioral problems are associated with a computerized performance-based assessment of attention deficit and hyperactivity disorders in rural school-aged African children.

## Impact statement

Because it has only 25 questions to which either a parent or teacher can respond in evaluating mental health problems exhibited by a child, the Strengths and Difficulties Questionnaire (SDQ) is often preferred as a screening tool in low- and middle-income countries. The SDQ is also freely available and easily accessible online with ample documentation for scoring in many different languages. However, it has not been used much in sub-Sahara Africa apart from its use with HIV-affected children in South Africa. Our study used the SDQ with mothers who have been part of a long-term follow-up study of their child's development, in gauging the benefits of a comprehensive health intervention treatment package for the mothers during pregnancy. Five hundred and sixty-six mothers whose study child had reached 6 years of age completed the SDQ for that child and the child also completed a computerized Tests of Variables of Attention (TOVA) assessment, providing a performance-based screening measure of attention deficit and hyperactivity disorders (ADHD). In this article, we investigate the associations between children's behavioral and emotional difficulties as measured by the SDQ, and the children's performance-based measures on a computerized TOVA administration. The TOVA was administered to the child in a separate room from the mother/caregiver, who responded to the spoken items from the SDQ presented by the interviewer in her primary language (usually *Fon*, spoken throughout Benin).

In correlating the SDQ to the TOVA assessments, we controlled for maternal depression, home outcomes measurement evaluation of quality of caregiving by the mothers, level of socioeconomic status (SES), the mother's educational level and the child's schooling status. TOVA measures of impulsivity were significantly related to SDQ emotional and hyperactivity/inattention difficulties. TOVA inattention was related to SDQ emotional difficulties. The maternal risk factors for which we controlled in these correlations also significantly predicted their child's SDQ total difficulties score. This is the first study we know of in Sub-Sahara Africa (SSA) that has effectively triangulated a child's mental health needs with parent report (SDQ), maternal risk factors (e.g., depression) and the child's performance on validated behavioral measures (ADHD). Combining such measures together can be much more effective in globally monitoring mental health needs from early to middle childhood, through adolescence.

## Introduction

The Strengths and Difficulties Questionnaire (SDQ) is one of the most widely used questionnaires that allow parents to report on their children's behavioral and emotional development (Goodman 1997). Other questionnaires such as the Achenbach Child Behavior Checklist and the Behavior Rating Inventory for Executive Function have been adapted and validated in the sub-Saharan African (SSA) context for parent- or teacher-based mental health or behavioral evaluation of preschool or school-aged children (Boivin et al. 2013a, b; 2020a; Ruiseñor-Escudero et al. 2015; Familiar et al. 2016; Familiar et al. 2015). In contrast to these instruments, however, the SDQ is shorter than most other well-validated mental health questionnaires (25 items), well validated in many clinical contexts and freely available in terms of both parent and teacher and self (adolescent) administration with scoring and other resources readily available online (youthinmind 2022). Because of these features, the SDQ is often considered the screening measure of choice for the early detection of behavioral and emotional problems in school-aged children and adolescents (Cianchetti 2020).

The SDQ contains questions about positive and negative child attributes across five items subdivided into five scales of five items each: emotional symptoms, conduct problems, hyperactivity/inattention symptoms, peer relationship problems and prosocial behavior. The emotional, conduct, hyperactivity/inattention and peer relationship scales are added together to generate total difficulties scores for the SDQ, while the prosocial behavior scale gives a positive or "strength-" based attributes measure (Goodman 1997).

SSA children may often be at risk for developmental delays and disorders accompanied by socioemotional symptoms and problems (Walker et al. 2007, 2011). This is because of the high prevalence of early exposures to infectious diseases affecting the brain and CNS very early in childhood, with little in the way of effective screening measures or interventions to prevent developmental delays and disabilities (Boivin et al. 2015). The developmental delays established early in child development in SSA children can then place them at greater risk for mental health issues in middle and late childhood (Davidson et al. 2015).

This makes the availability and validation of mental health screening measures such as the SDQ of vital importance in communities in SSA where children are especially at risk developmentally (Lovero et al. 2022). Such mental health screening is especially critical where the early developmental trajectory of these children is diminished due to impoverishment, limited resources, infectious disease, nutritional deprivation and the compounded effect of such risk factors within their developmental *milieu*, especially throughout the first 1,000 days (Worthman et al. 2016).

Toward that end, Hoosen et al. (2018a, b) published a scoping review of the application and validation of the SDQ in Africa (Hoosen et al. 2018a, b). They included 54 studies from 12 African countries in their review, with most being based in South Africa. Among these studies, the SDQ was typically used to evaluate internalizing and externalizing symptoms in clinical populations most often affected by HIV disease and/or exposure. The authors concluded that while useful as a mental health screening tool in the African context, few of these studies contributed to a better understanding of the psychometric and/or normative properties of the SDQ, and little was known about its validity in assessing non-HIV/AIDS-affected clinical samples for mental health problems (Hoosen et al. 2018b).

An exception to this scoping review conclusion was an application of the SDQ in Kinshasa, Congo, to evaluate its utility in screening for school children with ADHD problems (Kashala et al. 2006). In a sample of 357 school children between 7 and 9 years of age, the SDQ positively screened 183 children for ADHD symptoms, and this clinical subgroup tended to be associated with poorer school performance, greater conduct disorders as determined by teachers and younger maternal age at birth than the children that screened negative (Kashala et al. 2005, 2006).

Garrison and colleagues used the SDQ in rural 6-year-old children in Benin Africa, as evaluated by mothers who had been part of malaria and intestinal parasite and micronutrient supplement intervention program during pregnancy for those (Garrison et al. 2021). Higher SDQ internalizing symptoms (emotional and ADHD problems) were observed in children by mothers who themselves were diagnosed for soil-transmitted helminth infection during pregnancy. It should be noted, however, that Boivin and colleagues observed that such infections were also associated with significantly poorer socioeconomic status (SES) and living conditions in rural Congolese school-aged children (Boivin et al. 1993). In later studies by Garrison et al. (2022a) and (2022b) of this same maternal/child cohort, mothers with higher depressive symptoms also provided lower quality of care for these children as assessed by the home observation measurement of the environment (HOME) scale. Quality of caregiving have home mediated the relationship between poorer living situations and corresponding depression in these mothers, and poorer mental health outcomes on the SDQ as observed in the Garrison et al.'s (2021) study. Unfortunately, the results of the SDQ evaluation by the mothers for their children are not reported in Garrison et al. 2022a.

The main objective of the present research is to evaluate the relationship between the maternal report of their child's mental health using the SDQ at 6 years of age in this same Benin maternal/child cohort as reported in the Garrison et al.'s 2021 and 2022 studies. Furthermore, we will evaluate the relationship between the SDQ, and a performance-based computerized assessment of ADHD domains, using the Tests of Variables of Attention (TOVA) test. Boivin and colleagues validated the use of the TOVA in other clinical contexts in SSA (Boivin et al. 2018; Chernoff et al. 2018). So far as we know, this will be the first time that application of the SDQ in the clinical diagnosis of ADHD symptomology in school-aged children as reported by parent or teacher (Kashala et al.

2005, 2006), has been validated with a performance based screening measure of ADHD (i.e., the TOVA).

*The principal hypothesis of the present study is that performance on the TOVA will be strongly related to mothers' evaluation of their child's psychosocial difficulties as reported on the SDQ questionnaire.* More specifically, we hypothesize that overall, TOVA performance will be most strongly related to the SDQ domain of externalizing (behavioral) difficulties, comprised of the items on the Hyperactivity and Conduct scales. Furthermore, our present maternal/child cohort in Benin that is the basis of the present study is well characterized in terms of maternal risk factors such as depression, lower SES, poorer HOME caregiving quality or other such mediational or modifying factors for a child's mental health status as measured by the SDQ. Thus, we can control for these factors when evaluating the relationship between a parental-report measure of mental health (SDQ) and a performance-based measures of ADHD symptomology (TOVA) (Koura et al. 2013; Boivin et al. 2021).

## Methods

### Study design

The study population consisted of children born to pregnant mothers enrolled in a trial comparing the efficacy of sulfadoxine-pyrimethamine and mefloquine from 2009 to 2010 ($N = 1,027$ – see Figure 1). These two intermittent preventive treatments for malaria in pregnancy are being used in the Malaria in Pregnancy Preventive Alternative Drugs (MiPPAD) trial in Benin, West Africa. The inclusion of these women took place during their second trimester of pregnancy. They were followed until delivery and their children for up to a year after birth. It was at this age that the 747 children underwent their first neurodevelopmental assessment in the TOVI (Fon language meaning: Child from the country) study (Koura et al. 2013; Mireku et al. 2015a). At the age of 6, 580 of her children were reevaluated for neurocognitive development as part of the EXPLORE study (Boivin et al. 2021). This evaluation was carried out from June 2016 to October 2018 by psychologist and nurse investigators specifically trained in the use of the different scales made available to them in the Sékou and Attogon health centers. These studies have been described in more detail in other articles (Boivin et al. 2021; Garrison et al. 2021). Figure 1 is a flowchart depicting this follow-up assessment with the TOVA and SDQ at 6 years of age for our present study children.

### Assessment of child development

At 6 years of age, the study child and mother visited one of the two rural health centers about an hour north of the city of Cotonou (Attogon or Sékou). At the health center, trained investigators obtained written consent from the parents and then administered

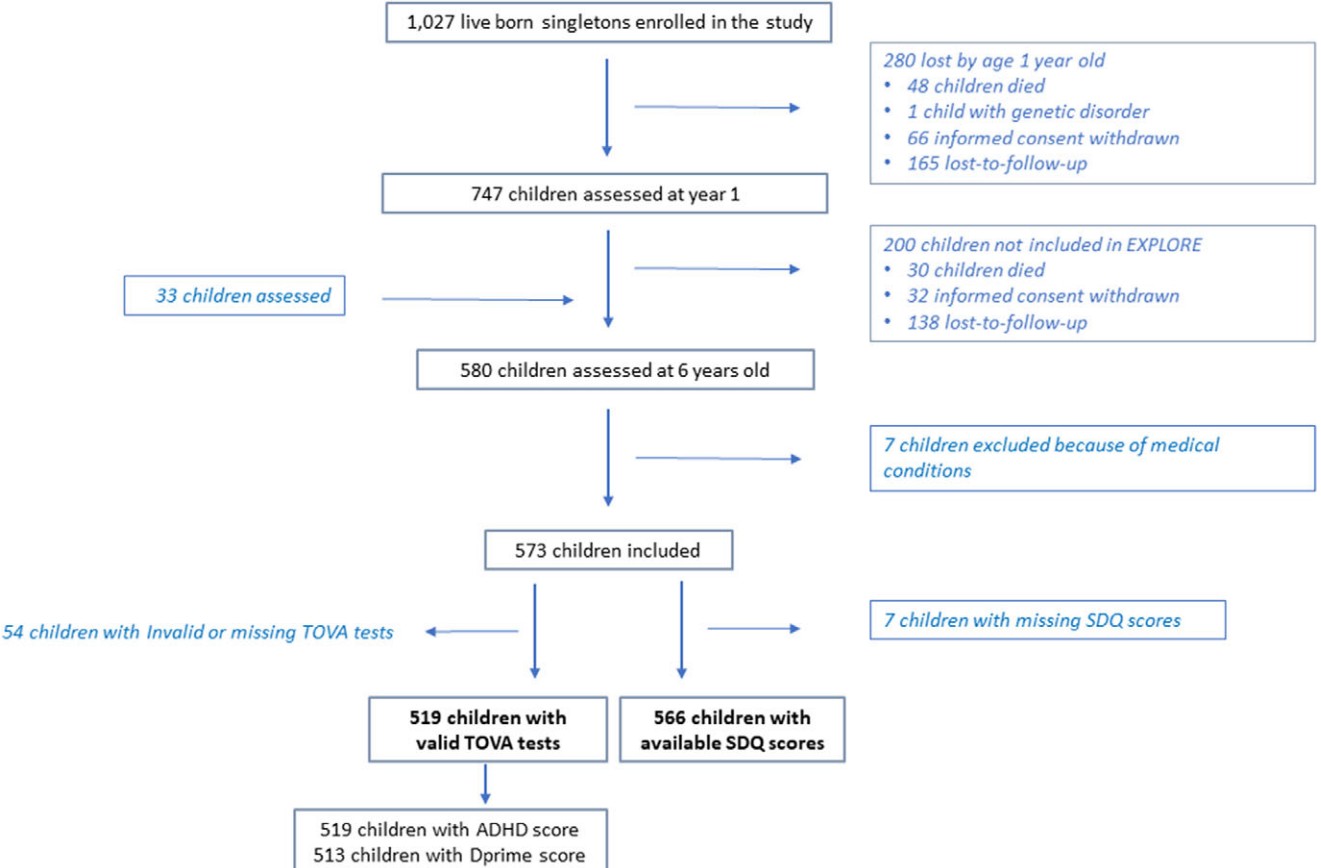

**Figure 1.** This flowchart depicts the initial enrollment of 1,027 live singletons born to mothers enrolled in the MiPADD malaria prevention during pregnancy prenatal care study. Seven hundred and forty-seven of these mother/child dyads were assessed a year after delivery as part of the TOVI study of child neurodevelopmental and health outcomes (Koura et al. 2013). From this cohort, 580 mother/child dyads participated in the EXPLORE follow-up assessment when the child was 6 years of age (Boivin et al. 2021). Five hundred and sixty-six mothers from this assessment completed the SDQ for the study child, and valid computerized TOVA tests were obtained from 519 of these children. Children having both maternal SDQ and valid TOVA measures were included in the present correlational analysis for this study.

questionnaires to the mothers and assessed the children's neuro-cognitive and behavioral development. If a child was sick or had a fever when they arrived at the clinic, blood samples were taken to determine the presence of an infection and the children were asked to return later for neurocognitive assessments after treatment. Children who tested positive for malaria, anemia or helminth infection were treated according to the national treatment guidelines for these conditions in Benin at the time and rescheduled for assessment in a few weeks. The maternal questionnaires and the child neuropsychological performance tests administered at this visit are recounted in more detail elsewhere (Boivin et al. 2021). This article focuses on the SDQ administered to the mother to evaluate the mental health of the study child, and the TOVA visual test administered to the study child on computer in a quiet assessment room at the clinic.

### Strengths and difficulties questionnaire

The SDQ is a 25-item screening questionnaire (Goodman, 1997) for assessing emotional and behavioral difficulties for children and youth (Mieloo et al. 2012). For younger children, either a parent or teacher can complete the questions, and the SDQ can be self-administered for older children (Goodman, 2001). In this study, the mother completed the assessment if she was the primary caregiver. The SDQ consists of 25 items divided into five subscales: hyperactivity/attention deficit; emotional problems; problems with peers; behavioral problems and prosocial behavior. The five sub-scales of five items each have a minimum score of 0 (lowest score) to 10 (highest score) and the first four subscales can be combined to obtain a total difficulty score, which can range from 0 to 40 points (youthinmind 2022). For the four difficulty scales and the total score, the higher the score the more difficulties the child is reported to have in that domain.

The five items of the prosocial scale provide a measure of strength for the SDQ, whereby the higher the score the better the social behavior of the child toward others. It can measure both problem behaviors and skills at an early age and has become one of the most widely used screening tools globally (Hoosen et al. 2018a, b). The SDQ differentiates between externalized emotional and behavioral problems in children (sum of the conduct and hyperactivity/inattention scales) and internalized (sum of the emotional problems scale and relationship with peers scale). Another advantage of the SDQ is that it is free and available online (www.sdqinfo.com). In disorder screening, only the first four scales are considered, and apart from the prosocial scale, the combined scale score reflects total difficulties, indicating the severity and content of emotional and behavioral problems (youthinmind 2022). The prosocial subscale indicates an inverse score and shows that the higher the scores, the more prosocial behaviors or characteristics the child has (Goodman 1997).

For this study, a common translation of the oral instructions from French to the Fon (national language of southern Benin) of the SDQ was validated by trained investigators (psychologists and nurses) before the start of the assessments. As most of the mothers were illiterate, all instructions were given to the mothers and guardians of the children by the investigators in Fon and orally (Garrison et al. 2021).

### Test of variables of attention

TOVA is a computerized visual continuous performance test used in to screen, diagnose and monitor children and adults at risk for ADHD (Greenberg 1993). TOVA (visual) consists of the rapid (tachistoscopic) presentation of a large geometric square on the computer screen with a smaller dark box either in the upper position (signal) or lower position (non-signal). The child is asked to press a switch held in the preferred hand as fast as possible in response to the signal (measuring vigilance attention), but to with-hold responding to the non-signal (measuring impulsivity). Following spoken instructions in the local language and practice trials, TOVA takes ~11 min for children 5–5.5 years old, and 22 min to administer for children 5.5 years and older. TOVA had been adapted for pediatric HIV research in Uganda (Boivin et al. 2010a, b, 2016; Ruel et al. 2012; Giordani et al. 2015) as well as for Ugandan school-aged children surviving cerebral malaria (Boivin et al. 2007; John et al. 2008).

The TOVA's primary outcome variables are response time variability (a sensitive indication of inattention), response time, percent commission errors (impulsivity), percent omission errors (inattention), an ADHD index score (missed signals in proportion to incorrect responses to non-signal) and the D-prime signal detection measure of overall test performance (correct signal "hits" in proportion to correct nonresponses to non-signal). For American standards, scores below $-1.80$ are suggestive of ADHD and higher ones denote normal levels of attention (www.tovatest.com) (Chernoff et al. 2018). The administration of the TOVA as part of the EXPLORE cohort in the present study has been previous described (Boivin et al. 2021). The TOVA instructions were spoken to the child in the local language of Fon, translated from the script of the instructions presented in English as part of the audio for the initial practice portion of the TOVA, to ensure that the child can respond correctly to the signal and non-signal stimuli before the start of the testing session.

### Socioeconomic status

We used two variables to assess family SES family wealth and maternal education when the child was assessed at 1 year of age. The family wealth scale has been described elsewhere (Koura et al. 2013). Briefly, it was assessed using a scoring instrument incorporating a checklist of material possessions (radio, television, bike, motorbike and car), possession of cows and access to electricity. Maternal education included as schooled or unschooled at the primary school level at least.

### Maternal level of formal education

The mother's educational level was categorized as either No Schooling, Partial Primary Schooling, Primary Schooling (grade 6) Complete or Secondary Schooling and beyond. None of the mothers in the present study sample had completed university.

### Child enrolled in primary school

This measure was categorized as either yes or not, since the children were almost all 6 years of age at the time of assessment, and none had progressed beyond grade 2 if enrolled in school.

### Home Observation for Measurement of the Environment

The Home Observation for Measurement of the Environment (HOME) (Caldwell & Bradley, 1979) was administered. This evaluation of the quality of the developmental environment and quality of caregiving by the mother was done in the home of the children and their mother when the child was assessed at 1 year of age, when it was adapted and piloted for this setting (Koura et al. 2013).

### Edinburgh postnatal depression scale

This maternal questionnaire has been used to assess depressive symptoms when the child was assessed at 1 year of age (Cox et al. 1987; Hanlon et al. 2008; Kakyo et al. 2012). Scores derived from the Edinburgh Postnatal Depression Scale (EPDS) were analyzed as a continuous variable. The EPDS was already available in French, but any spoken instructions to the mother were in Fon by our study psychologist and several nurses, in that they knew the local dialect spoken my mothers in at our two study sites (Koura et al. 2013).

### Statistical analysis

Maternal and child measures were summarized with means and standard deviations for continuous variables (e.g., SDQ, TOVA, SES, EPDS and HOME) and counts and percentage for categorical variables (e.g., maternal educational level, child enrolled in school). Composite cognition scores and gross motor scores from the SDQ were related to TOVA performance domains using regression analysis. Following unadjusted analyses with one predictor (SDQ), multivariable general linear models were fit controlling for maternal sociodemographic measures (SES, education level and depression level), child's gender, whether enrolled in school, and HOME total score for the child's quality of caregiving and developmental milieu based on evaluation during a visit to the child's home. These factors were selected *a priori* based on subject matter expertise and literature documenting these factors as potentially important for neuropsychological outcomes. Based on this modeling analysis, correlations between the principal measures from the TOVA and the SDQ were adjusted for maternal education, child enrolled in school or not, SES (material possessions score), Caldwell HOME scale total score and EPDS total score.

The resulting Pearson partial correlation coefficients computed between a TOVA and an SDQ measures evaluated the strength of the relationship (Table 3), is equivalent to the Pearson correlation between the residuals of the two variables after regression on the controlling variables. This analysis uses three stages to calculate the partial correlation between variables X1 and X2, while controlling for the covariate measures Z1, Z2 and Z3: 1) Linear regression or ANOVA analyses of X1 (dependent variable) on all the Z variables (independent variables). The same for X2. For the next stage, residuals of both regressions are derived, and a Pearson correlation is calculated (*r*).

Finally, *p* value is computed by treating $\frac{(n-k-2)^{(1/2)}r}{(1-r^2)^{1/2}}$ as coming from a Student distribution with $(n - k - 2)$ degrees of freedom, where *r* is the partial correlation and *k* is the number of variables being partialled out. The resulting Pearson product–moment correlation coefficients are presented in Table 3, which is the principal table statistically evaluating the relationship between the TOVA and SDQ measures (after partialling out the study covariates). All statistical tests were two-sided at a significance level of .05. All the analyses were performed with SAS 9.4 software. SAS 9.4 was also used to generate the histogram/box plots in Figure 2. However, the scatterplot arrays for the TOVA and SDQ measures in Figure 3 were generated using the statistical program R (Team, 2023).

## Results

### Sociodemographic characteristics

Of the 1,027 children born from singleton births and born alive to mothers in the MiPPAD clinical trial, 580 children still had their biological mother as their principal caregiver and were assessed for neurocognitive and behavioral development at age 6 as part of the EXPLORE project. Age data for three children were missing and were excluded from the analyses, as well as for two other children, one of whom was less than 5 years of age, and the other was greater than 8 years of age (Figure 1). The study samples consisted of 571 mother/child dyads, of whom 566 mothers completed the SDQ questionnaire for the study child and 519 children completed a valid computerized TOVA evaluation (Table 1). These were almost evenly divided by gender (boys and girls) and descriptive information for the children in this study who completed these assessments is included in Table 1. None of the descriptive measures for the TOVA or SDQ samples in Table 1 differed from the entire sample (All) eligible for consideration in the present analysis in terms of a statistical comparison between the TOVA or SDQ sample and the entire sample.

### Description of SDQ and TOVA measures for present sample

Table 2 presents the histogram for the distribution of the two TOVA measures representing overall performance on this test: the ADHD index score and the D-prime signal detection score. In the far-right hand column for both of these two TOVA principal performance measures, the first (Q1) and third (Q3) quartile values are included along with the median (Q2). The minimum and maximum scores are also included. Of the 566 children whose mothers completed the SDQ and the 519 children able to complete the TOVA because the availability of electrical power for the desktop computer and monitor used for administration and/or completion of the testing session without outside interruption or power failure. Five hundred and seven children had complete and valid scores on all the measures for both the SDQ and the TOVA. It is these children ($N = 507$) who are included in the principal analyses evaluating the relationships among these two assessments.

Although in Table 2, some of the standardized measures (using US norms) are low (i.e., D-prime standard score < mu = 100), the distribution of continuous scores for all the TOVA performance domains in the lower portion of Figure 2 are reasonably unimodal with no clear basement or ceiling effect. This is depicted for the ADHD index score and the D-prime standardized scores in the bar graphs contained in this figure as well. The ADHD index score is computed based on US gender and age performance measures and can range from −10.0 to +10.0 with a mean at about 0. The ADHD index performance is unimodal, but below the 0 mean. Likewise, the unimodal cluster for the D-prime standard score well below the normative mean of 100, although the US norms are not representative for sub-Saharan populations for the TOVA (Boivin et al. 2007, 2020b; Boivin and Giordani, 2009; Bangirana et al. 2013).

### Association of TOVA measures with SDQ parent report

Table 3 presents the partial Pearson product–moment correlation coefficients for all the TOVA performance outcomes: ADHD index, D-prime signal detection score, percent commission errors, percent omission errors, response time to signal and response time variability to signal. These are correlated with the scale scores for the SDQ (conduct, hyperactivity, emotions and peer relationships) for difficulties, along with the prosocial scale score. The SDQ total difficulties score is also included along with total externalizing (behavior) problems (conduct and hyperactivity scales) and internalizing (emotional) problems (emotions and peer relationships scales). The SDQ prosocial scale is a stand-alone measure of

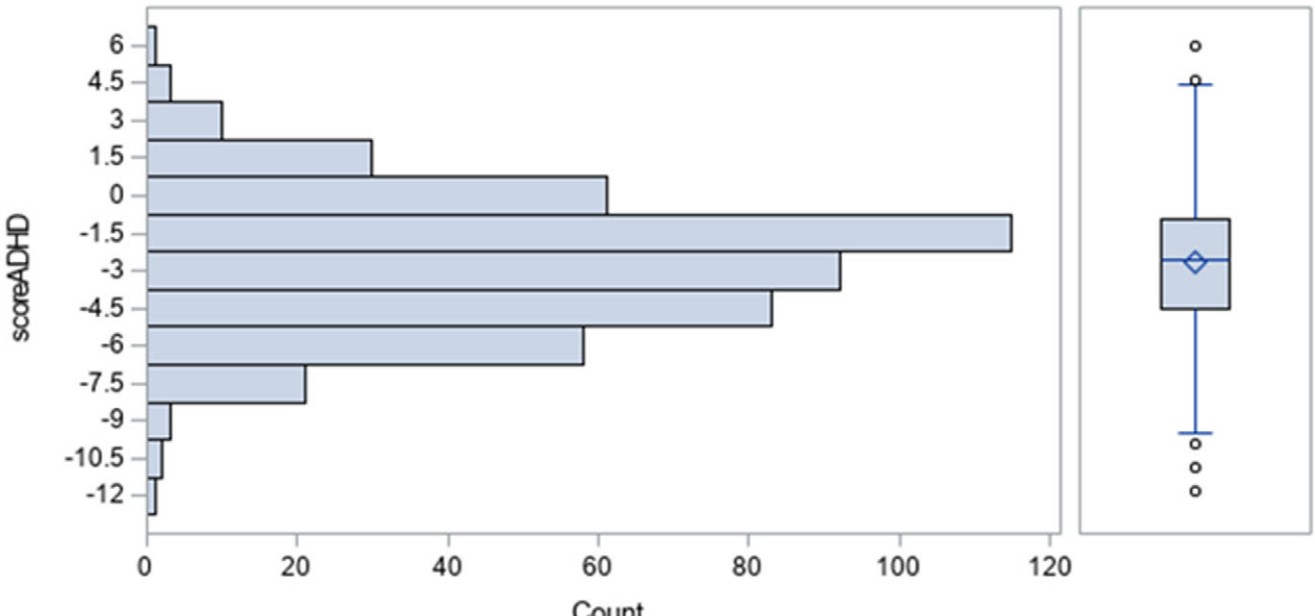

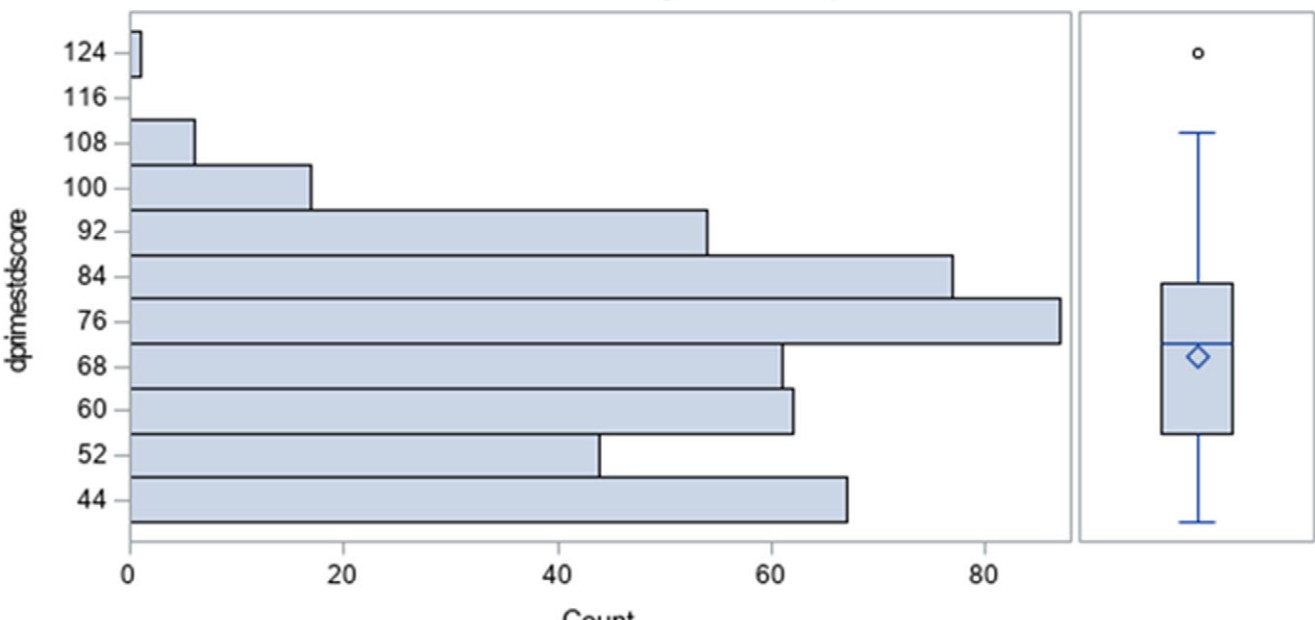

**Figure 2.** The upper portion of this figure shows the histogram distribution of ADHD index scores for the Tests of Variables of Attention (TOVA) assessment for the present sample of children. To the far right is the box plot depicting the median (bar), first and third quartiles (bottom and top of box), and range of scores excluding the outliers (depicted individually as data points above and below the box plot range). Similar information is depicted in the bottom portion of this graph for the TOVA D-prime signal detection measure for correct response to signal, standardized based on American norms by age and gender.

positive emotions and behaviors toward others, representing the "strengths" portion of the SDQ.

The TOVA measure of percent errors of commission, which indicates impulsivity as children response to the non-signal as if it were the signal, was the performance measure most strongly related to the SDQ. A partial correlation coefficient was obtained for the SDQ total difficulties ($r_\mathrm{p}$ = 0.14(506), $p$ = 0.001), externalizing difficulties ($r_\mathrm{p}$ = 0.13(506), $p$ = 0.004) and internalizing difficulties ($r_\mathrm{p}$ = 0.10(506), $p$ = 0.024) measures (Table 3). The relationship

between this TOVA measures of impulsivity was especially strong with the SDQ scales of hyperactivity difficulties ($r_\mathrm{p}$ = 0.117(506), $p$ = 0.009) and emotional difficulties ($r_\mathrm{p}$ = 0.125(506), $p$ = 0.005). Other TOVA measures significantly related to SDQ total difficulties were D-prime ($r_\mathrm{p}$ = −0.103(506), $p$ = 0.02) and correct response time variability ($r_\mathrm{p}$ = 0.125(506), $p$ = 0.005), with the latter being a sensitive measure of inattention. This TOVA measure of inattention was especially correlated with internalizing difficulties ($r_\mathrm{p}$ = 0.114(506), $p$ = 0.01), emotional difficulties ($r_\mathrm{p}$ = 0.114

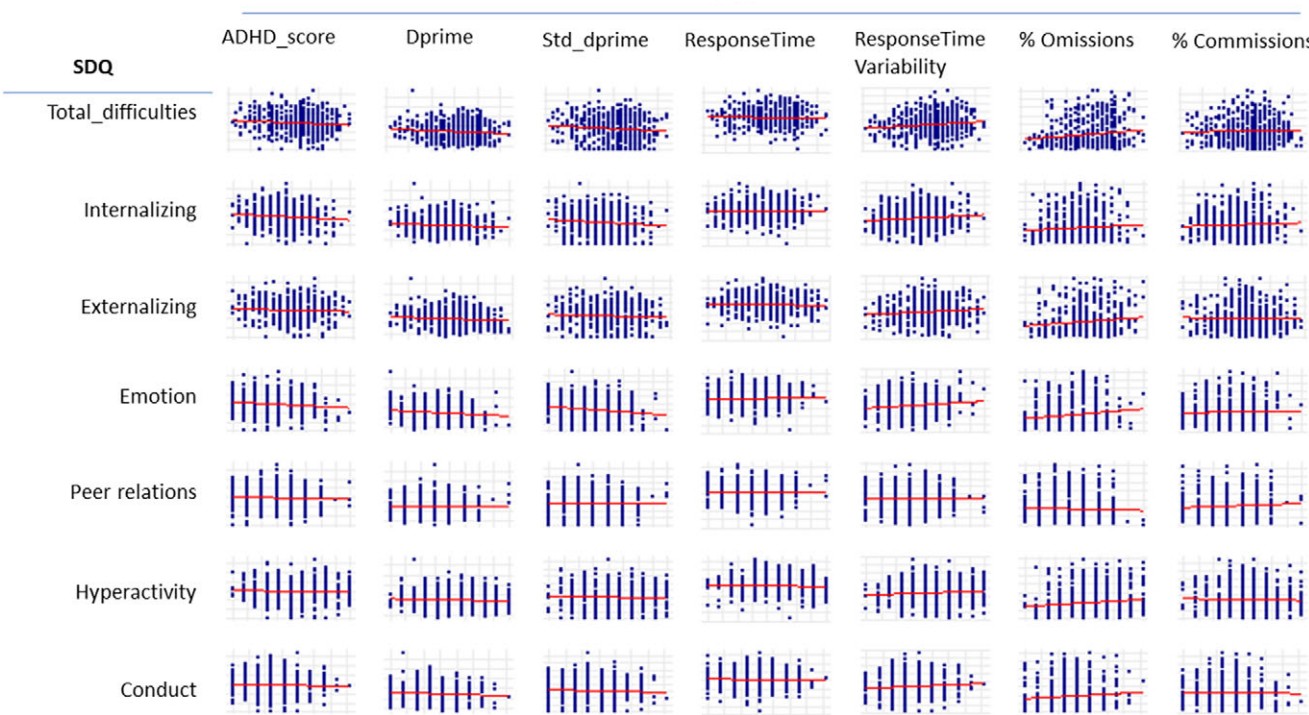

**Figure 3.** These scatterplots present by row the principal Tests of Variables of Attention (TOVA) performance measures (ADHD index score, D-prime signal detection score, D-prime standardized signal detection score, average response time to signal, response time variability to signal, percent omission errors to signal [inattention], percent commission errors to non-signal [impulsivity]). Each of these TOVA outcomes is plotted against the principal composite sums for the Strengths and Difficulties Questionnaire (SDQ) assessment. The SDQ rows from top to bottom are for SDQ Total difficulties, SDQ Internalizing difficulties (Emotion and Peer relations scales) and SDQ Externalizing difficulties (Hyperactivity and Conduct scales). Each scatterplot includes a linear least square fitted line (red), whereby the slope of the line visually represents the strength of the relationship between that TOVA (X axis) and SDQ (Y axis) measure.

(506), $p = 0.01$) and the prosocial scale of the SDQ ($r_p = 0.10(506)$, $p = 0.03$) (Table 3). Figure 3 is comprised of scatterplots between the principal TOVA performance measures and the principal SDQ difficulty measures with significant partial correlation coefficients in Table 3. The scatterplot toward the lower left corner of Figure 2 depicts a strong linear positive trend between TOVA percent commission errors and SDQ total difficulties, as is evident from the partial correlation coefficient for these two measures in Table 3. The more difficulties as reported by the parent on the SDQ, the greater the number of impulsive errors (percent commission errors to non-signal presentation) on the TOVA (positive correlation).

## Discussion

The principal hypothesis of the present study was that performance on the TOVA would be strongly related to mothers' evaluation of their child's psychosocial difficulties as reported on the SDQ questionnaire. The hypothesis was supported in that there were several statistically significant Pearson product–moment partial correlation coefficients between SDQ (total difficulties, externalizing difficulties and internalizing difficulties) and TOVA measures of performance in terms of impulsivity (percent commission errors to non-signal) and inattention (D-prime signal detection, and correct response time variability to signal) (Table 3). More specifically, however, we hypothesized that TOVA performance would be most strongly related to the SDQ domain of externalizing (behavioral) difficulties, comprised of the items on the hyperactivity and conduct scales. This is because of the TOVA assessment focus on the twin

ADHD domains of inattention and impulsivity. In our present findings, whereas the TOVA impulsivity measure (percent commission errors) was most strongly correlated with the SDQ externalizing difficulties scales, the inattention measures were not (e.g., percent omission errors, correct response time variability, D-prime signal detection). Those TOVA measures were more strongly related to the SDQ internalizing (emotional) difficulties scales (Table 3).

Even though, the few if any of the rural children at 6 years of age in the present cohort had any prior experience with a laptop computer either in the home or at school, the vast majority (519 out of 571 or 91%) were able to complete the computerized TOVA test with valid performance measures pertaining to ADHD. In the present study, the TOVA should not be used diagnostically for ADHD, but as a performance-based measure of ADHD domains to relate to the SDQ obtained from parents (caregiver report mental health screening measure). This is because the overall performance means on the ADHD index and the D-prime standardized signal detection measures for our present sample were well below the expected American-based norms. However, our TOVA test still proved useful as a performance-based measure of ADHD propensities when used as an evaluative research tool strictly within our present cultural and sociodemographic cultural context.

Likewise, their mothers or principal caregivers were able to respond to the SDQ questionnaire when the questions were spoken to them in their local language, to provide a mental health evaluation of their child (566 out of 571 or 99%). The SDQ and the TOVA can be used with school-aged children in such resource-constrained SSA settings to screen for emotional and behavioral problems using both parent-based report and child-based

**Table 1.** Characteristics of the study population children completing Tests of Variables of Attention (TOVA) assessment and for mothers completing the Strengths and Difficulties Questionnaire (SDQ) for their study child

| Descriptive characteristic of study sample | TOVA sample N = 519 | SDQ sample N = 566 | All N = 571 |
|---|---|---|---|
| Child's sex | | | |
| Male | 261 (50.3) | 283 (50) | 285 (49.9) |
| Female | 258 (49.7) | 283 (50) | 286 (50.1) |
| Child age in years at TOVA/SDQ tests | 6.3 (0.5) | 6.3 (0.4) | 6.3 (0.4) |
| Child in school at the age of TOVA/SDQ tests | | | |
| Yes | 336 (65.5) | 379 (67.1) | 379 (67.1) |
| No | 177 (34.5) | 186 (32.9) | 186 (32.9) |
| Missing | 6 | 6 | 6 |
| Maternal marital status | | | |
| Single/divorced/widowed | 29 (5.7) | 33 (5.9) | 33 (5.9) |
| Married (monogamous) | 243 (47.8) | 271 (48.4) | 271 (48.4) |
| Married (polygamous) | 236 (46.5) | 256 (45.7) | 256 (45.7) |
| Missing | 11 | 11 | 11 |
| Maternal education | | | |
| No schooling | 338 (66.4) | 359 (64) | 359 (64) |
| Partial primary schooling | 101 (19.8) | 121 (21.6) | 121 (21.6) |
| Primary schooling complete | 19 (3.7) | 21 (3.7) | 21 (3.7) |
| Secondary schooling and more | 51 (10) | 60 (10.7) | 60 (10.7) |
| Missing | 10 | 10 | 10 |
| Socioeconomic status (SES) score | 5.8 (2.9) | 5.9 (2.9) | 5.9 (2.9) |
| Missing | 8 | 8 | 8 |
| Maternal Edinburgh Postnatal Depression Scale (EPDS) | 4 (5.4) | 4.3 (5.4) | 4.3 (5.4) |
| Missing | 16 | 16 | 16 |
| **Caldwell Home Observation for Measurement of the Environment (HOME) Inventory** | 26.9 (2.3) | 27 (2.2) | 26.9 (2.2) |
| Missing | 29 | 29 | 29 |

performance, respectively. The feasibility of such measures further supports the adaptation of other such "western-based" tests that have been previously adapted by our group in the neurodevelopmental and neurocognitive evaluation of this rural Benin cohort; allowing us to gauge the impact of such outcomes in response to perinatal and related maternal risk factors in the first 1,000 days of SSA child development (Mireku et al. 2015b, 2016; Boivin et al. 2019, 2021; Garrison et al. 2021, 2022a).

The range and unimodal descriptive properties of our SDQ scale and TOVA performance measures without ceiling or basement effects from our present sample (Table 2 and Figure 2) also supports the utility of these screening measures for mental health and behavioral problems in our present SSA context. Such distributive features were foundational to the statistically significant and clinically meaningful partial correlation coefficients observed in our present findings (Table 3). Even when controlling for maternal education, child schooling, SES, HOME quality of caregiving and EPDS maternal depression – significant relationships were evident between the TOVA measure of behavioral impulsivity (percent commission errors in response to non-signal) and almost

all the SDQ scales (conduct, hyperactivity and emotional difficulties and problems). TOVA impulsivity (commission errors) was also significantly correlated with the SDQ composite indices (total difficulties, externalizing problems and internalizing problems) (see Table 3).

The TOVA vigilance attention measures of D-prime (signal detection measure of response to signal) and of response-time variability to the signal (measure of inattention) were significantly correlated with the SDQ internalizing (emotional symptoms) measure. Together, these findings seemed to indicate that in the present study, TOVA inattention corresponded to emotional problems while impulsivity corresponded to behavior problems in the SDQ parent screening questionnaire. In contrast, the ADHD index for overall risk for ADHD on the TOVA was correlated with the SDQ prosocial scale in the expected direction (greater risk for ADHD on the TOVA, lower SDQ Prosocial Scale score). Again, this is the first time that the SDQ has been evaluated in terms of its correspondence to a performance-based test for ADHD so far as we know. Again, such correspondence validity between the TOVA and SDQ was evidenced in a rural SSA context with younger school-

**Table 2.** Distribution of the Strengths and Difficulties (SDQ) – parent response, and Tests of Variables of Attention (TOVA-visual) scores

| | N | Minimum | Q1 | Median | Q3 | Maximum | Interpretation |
|---|---|---|---|---|---|---|---|
| Strengths and Difficulties Questionnaire (SDQ) – Parent completion for study child (N = 566) | | | | | | | |
| Total SDQ difficulties | 566 | 8.0 | 19.0 | 23.0 | 25.0 | 33.0 | |
| Internalizing difficulties score | 566 | 0.0 | 4.0 | 7.0 | 9.0 | 15.0 | |
| Emotional difficulties | 566 | 0.0 | 2.0 | 4.0 | 5.0 | 10.0 | |
| Peer relationship difficulties | 566 | 0.0 | 2.0 | 3.0 | 4.0 | 8.0 | |
| Externalizing difficulties score | 566 | 0.0 | 7.0 | 9.0 | 11.0 | 17.0 | |
| Conduct difficulties | 566 | 0.0 | 2.0 | 3.0 | 5.0 | 9.0 | |
| Hyperactivity difficulties | 566 | 0.0 | 4.0 | 5.0 | 7.0 | 10.0 | |
| Prosocial strengths | 566 | 0.0 | 5.0 | 6.0 | 8.0 | 10.0 | |
| Tests of Variables of Attention (TOVA-visual) test (N = 519) | | | | | | | |
| ADHD index score (adjusted by age/gender) | 519 | −10.0 | −4.5 | −2.5 | −0.8 | 6.0 | Attention deficit hyperactivity disorders performance index |
| D-prime signal detection | 513 | −0.4 | 0.7 | 1.5 | 2.2 | 5.6 | Correct responses to signal and non-signal proportional to incorrect responses |
| D-prime score (standardized by US norms) | 519 | 40.0 | 57.0 | 72.0 | 84.0 | 124.0 | Correct responses to signal and non-signal proportional to incorrect responses (standard score) |
| Percent error of commission | 519 | 0.2 | 8.3 | 15.4 | 34.9 | 86.4 | Measure of impulsivity (response to non-signal) |
| Percent error of omission | 517 | 0.9 | 15.4 | 26.5 | 42.0 | 92.0 | Measure of vigilance attention no response to signal) |
| Correct mean response time to signal (ms) | 519 | 292.0 | 684.0 | 764.0 | 855.0 | 1,150.0 | Measure of speed of responding and the reactivity |
| Correct response time to signal variability (ms) | 519 | 145.0 | 269.0 | 313.0 | 370.0 | 532.0 | Measure of consistency in the speed of responding (attention measure) |

Minimum, first quartile (Q1), median (Q2), third quartile (Q3) and maximum score are presented along with a brief description/interpretation of the TOVA measures

aged children while controlling for maternal risk factors of influence for both sets of outcomes, in a well-characterized mother/child cohort that has been longitudinally evaluated (Koura et al. 2013; Boivin et al. 2021; Garrison et al. 2021).

We also observed important relationships when modeling the significance of how well our covariate measures (maternal educational level, not-at-school child, SES, HOME, Maternal depression [EPDS]) predicted our SDQ composite indices (total difficulties, internalizing problems and externalizing problems). SDQ scale measures of emotional symptoms and hyperactivity/inattention difficulties were especially well predicted by these covariates in our present sample. This finding was consistent with the important of maternal depression and SES for predicting SDQ emotional problems as observed by Garrison and colleagues in this same cohort (Garrison et al. 2021). Kashala et al. (2005, 2006) likewise when using the SDQ to identify Congolese school-aged children at risk for ADHD type problems, observed younger maternal age at birth and greater difficulties in school-related performance and conduct than non-ADHD community controls.

Strengths of this present study include a larger sample size of boys and girls (more than 500 dyads) followed longitudinally from birth through 6 years of age, who were well characterized in terms of medical and gestational history. The study was innovative in that a performance-based measure of ADHD that was cross-culturally adaptable and used in other rural and urban settings with at-risk populations in the SSA, was related to a widely used screening measure of psychosocial difficulties (SDQ) also used extensively in the SSA. Caregiver report-based screening measures of psychosocial, neurocognitive and other developmental difficulties are rarely related to performance or clinically based evaluative measures for these domains. Ours is the first such study in the SSA of which we are aware. Furthermore, because our present study sample was part of a well-characterized longitudinal cohort that has been followed-up to the present time, we were able to control for (partial out) various maternal and caregiving factors that may have influenced the TOVA or the SDQ as part of our correlational analysis. In fact, the present study cohort, now aged 13–14 years, is undergoing another comprehensive psychosocial and neurocognitive assessment, the results of which we hope to make available in the coming year or two.

## Strengths and limitations

However, a study limitation was the fact that by 6 years of age, only about half of the original dyads for the antenatal malaria prevention intervention were still available of evaluative follow-up with the SDQ and TOVA, in that the biological mother was still the principal caregiver. We do not know the extent to which the strong correlations we observed between the SDQ and TOVA would extend to children no longer able to caregiving situations for the child where the biological mother was no longer present. Likewise, the extent to those lost to follow-up would have presented with similar SDQ and TOVA correlational findings. Finally, the TOVA is a computer-based performance-based measure of ADHD-type difficulties and is proprietary and can be costly for clinical application for resource-constrained settings (US$15 per TOVA test credit – see www.tovatest.com). Also, the TOVA necessitates a quiet and private room for valid administration, and this can be difficult to

**Table 3.** Partial Pearson product–moment partial correlation coefficients between SDQ and TOVA measure; adjusted for maternal education, child enrolled in school or not, socioeconomic status (material possessions score), Caldwell Home Observation for Measurement of the Environment (HOME) scale and Edinburgh Postnatal Depression Scale

| Tests of Variables of Attentions (TOVA – visual) | Strengths and Difficulties Questionnaire (SDQ) – Parent Response for Study Child | | | | | | | |
|---|---|---|---|---|---|---|---|---|
| | Difficulties total | Externalizing diff. scales | Internalizing diff. scales | Conduct problems | Hyperactivity inattention | Emotional symptoms | Peer difficulties | Prosocial scale |
| ADHD index score (US norms) | −0.084 | −0.042 | −0.097 | −0.029 | −0.039 | −0.081 | −0.072 | −0.089 |
| *p* value | .06 | .35 | .03* | .52 | .39 | .07 | .11 | .046* |
| D-prime score – signal detection | −0.103 | −0.068 | −0.100 | −0.064 | −0.051 | −0.093 | −0.061 | −0.040 |
| *p* value | .02* | .13 | .03* | .16 | .26 | .04* | .18 | .38 |
| D-prime score (standardized US norms) | −0.101 | −0.063 | −0.103 | −0.048 | −0.055 | −0.101 | −0.057 | −0.053 |
| *p* value | .02* | .16 | .02* | .28 | .22 | .02* | .20 | .23 |
| Percent error of commission | 0.14 | 0.129 | 0.101 | 0.091 | 0.117 | 0.125 | 0.024 | 0.028 |
| *p* value | .001*** | .004** | .024* | .04* | .009** | .005** | .59 | .53 |
| Percent error of omission | 0.020 | −0.027 | 0.065 | −0.008 | −0.034 | 0.012 | 0.101 | .037 |
| *p* value | .66 | .54 | .15 | .86 | .45 | .80 | .02* | .41 |
| Correct response time to signal (ms) | 0.003 | −0.038 | 0.048 | −0.024 | −0.036 | 0.029 | 0.050 | 0.097 |
| *p* value | .95 | .40 | .28 | .59 | .42 | .51 | .27 | .03* |
| Correct response time variability (ms) | 0.125 | 0.088 | 0.114 | 0.072 | 0.073 | 0.106 | 0.071 | 0.100 |
| *p* value | .005** | .049* | .01** | .11 | .10 | .02* | .11 | .03* |

Note: *$p$ < .05, **$p$ < .01 and ***$p$ < .001, denoting statistically significant thresholds. *N* = 507 for sample of children with both SDQ and TOVA measures available.

arrange in crowded and busy outpatient care settings in either village clinics or large urban public hospitals in a comprehensive pediatric care setting. Finally, both the SDQ and the TOVA require the training of healthcare workers for proper administration, scoring and interpretation of scored findings. These requirements limit the accessibility to this performance-based measure, should a child screen positive for behavioral and emotional difficulties on the SDQ and further follow-up is recommended with such culture-adaptable tests previously validated in the SSA, such as the TOVA.

## Conclusion

The principal hypothesis of the present study was that performance on the TOVA would be strongly related to mothers' evaluation of their child's psychosocial difficulties as reported on the SDQ questionnaire. Overall, our correlational findings supported this hypothesis. Although the TOVA is designed to evaluate comprehensively screen ADHD problems using a performance-based and culturally adaptable computer-based measure, only the "impulsivity" domain (i.e., percent commission errors to non-signal) of the ADHD was significantly related to the hyperactivity scale of the SDQ. However, other attention-deficit TOVA measures were significantly related to the internalizing (emotional) domain, suggesting that internalizing and externalizing difficulties may have a good deal of overlap when it comes to the array of emotional and behavioral difficulties encompassed in psychosocial screening based on caregiver report (SDQ). We are planning a follow-up paper to this one which will examine the psychometric properties and factor structure of the SDQ with our rural SSA study population in much greater detail, to see how well differentiated internalizing and externalizing difficulties are for the scales to which they apply.

Because for the present study, we included only those mother/child dyads where the primary caregiver was still the biological mother, our findings may not extend to caregiver/child dyads where the primary caregiver responding to the SDQ items for the child is not the biological mother. Likewise, the relationship between maternal risk factors in the first 1,000 days of life for the child and the SDQ and/or TOVA outcomes for the relevant child at school age may not be as clearly apparent in households where the primary caregiver is no longer the study child. However, our findings do strongly suggest that the combination of caregiver report (SDQ) with a performance-based assessment of behavioral and social problems in vulnerable SSA child populations is feasible and worthwhile.

Furthermore, such screening assessments to identify such children especially at-risk for mental health issues are an important dimension of pediatric care and the educational needs for such children. As established by Boivin and colleagues in other SSA contexts with school-aged children, mental health and behavioral outcomes provide by such performance-based and caregiver-report measures can be of vital importance in establishing the evidence-based interventions in preventing developmental trajectories in early childhood resulting in developmental delays and mental health difficulties (Boivin et al. 2013a, b; Barros and Ewerling, 2016; Bass et al. 2016, 2017). As such children age into adolescence in SSA, the SDQ as a tool for the self-report of mental health problems can be scaled up for community-wide screening (Lovero et al. 2022). Likewise, the present study has established the feasibility of the TOVA as a performance-based measure to

evidence the benefits of caregiver-training interventions with the mother (Boivin et al. 2020a). Triangulating a child's mental health needs with parent report, self-report and performance on validated behavioral measures, can be of great value in continuing to monitor for ongoing mental health treatment needs across the child's life-span from adolescence into adulthood (Davidson et al. 2015).

**Open peer review.** To view the open peer review materials for this article, please visit http://doi.org/10.1017/gmh.2024.128.

**Data availability statement.** Data are available from the senior author of the study (F.B-L.) upon reasonable request.

**Acknowledgements.** The authors would like to thank the entire staff of the three health centers (Allada, Attogon and Sékou) just north of Cotonou, Benin. The authors would also thank the study participants at those centers.

**Author contribution.** F.B-L. and R.Z. conceived the study. L.C. and J.W. supported data collection. N.C. analyzed the data supervised by F.B-L. and M.J.B. and R.Z. were the principal authors in drafting this manuscript, supervised by F.B-L. All authors contributed to the interpretation, editing of the paper and approval of the paper for submission.

**Financial support.** Data collection costs were supported by funding from the Fondation de France, France (00100075) to F.B-L.

**Competing interest.** The authors declare no conflict of interest.

**Ethics standard.** All authors declare to adhere to the publishing ethics of Global Mental Health. Permission for the Human Subjects Research Ethical collection of data protocol was approved by the University of Abomey-Calavi's institutional review board (IRB), the Committee of Ethical Research of the Sciences Institute (CER-ISBA) and the French Institut de Recherche pour le Développement's (IRD) Consultative Ethics Committee. The consent process was conducted in the local language for all study women and those who participated in this study signed informed consent before enrollment or provided a thumbprint endorsed by a witness if the mother was unable to read or write.

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
