## [Reviewer Report]

This is a well written study that provides valuable information on a behavioral rating scale and an objective cognitive test in a unique community sample in SSA. Findings highlight the overlap be

---

## [Reviewer Report]

Thank you for the opportunity to review this manuscript. The study has the potential to make a valuable contribution to its field, particularly by providing unique insights from a rural context in Sub-Saharan Africa. However, there are significant issues that need to be addressed before the manuscript is suitable for publication.

First, I strongly recommend the authors to adhere strictly to STARD guidelines (https://www.equator-network.org/wp-content/uploads/2015/03/STARD-2015-checklist.pdf). Second, there are several key concerns.

1. Lack of hypotheses: The study’s objective is unclear due to the absence of stated hypotheses. Clearly defining hypotheses would clarify the study’s purpose and direction.

2. Results: P-values should never be reported without accompanying estimates. It is crucial that the authors consider these estimates when interpretating the results. Additionally, the rationale behind including a ROC analysis in the study is unclear and should be clarified.

3. Conclusions: The conclusions are undermined by the lack of predefined hypotheses. If the study aimed to assess the utility of TOVA, a logical hypothesis might be that TOVA scores would show the strongest associations with hyperactivity subscores. However, the findings indicate minimal correlation between TOVA scores and hyperactivity subscales, suggesting that these Western methods may not be well-suited for this population. This raises questions about the approach and whether this should be reconsidered.

4. Strengthen the manuscript’s structure: There is a need to strengthen the formal structure of the manuscript. Information that belongs to the “Methods” section is currently in the “Introduction”, and some of the results are present in the “Methods” section. Additionally, the “Results” section contains information that should be in the “Discussion”. Furthermore, the discussion needs a clear “Strengths and limitations” section. Tidying up and tightening the structure will enhance clarity.

5. Missing flowchart: On page 6, under “Study Design”, the authors refer to a flowchart (Figure 1). However, the flowchart is missing; Figure 1 in the manuscript is currently a histogram. In addition, preferably the flowchart will be presented in “Results section”

6. Inconsistent participant numbers: The abstract states that 561 mothers completed the SDQ, but the “Impact Statement” and “Results” sections mention 566. Please ensure consistency in these numbers.

Overall, based on the points mentioned above, I must respectfully recommend the rejection of the manuscript

---

## [Editor Report]

Can you kindly address the reviewers' comments, particularly tiding the manuscript, so that information is placed correctly to ensure clarity. Importantly, utilise a standardised research reporting checklist to ensure that all essential information is correctly placed and reported.